# The Influence of Smartphone Use on Tweens’ Capacity for Complex Critical Thinking

**DOI:** 10.3390/children10040698

**Published:** 2023-04-08

**Authors:** Rosa Angela Fabio, Rossella Suriano

**Affiliations:** 1Department of Economy, University of Messina, Via dei Verdi, 75, 98122 Messina, Italy; 2Department of Cognitive, Psychological and Pedagogical Sciences and Cultural Studies, University of Messina, 98100 Messina, Italy

**Keywords:** critical thinking, problematic smartphone use, tweens

## Abstract

The spread of modern technologies exposes many people to a high level of ambiguous and misleading information that can impact people’s judgments and worldviews. This happens especially in a phase of life such as preadolescence when children are particularly sensitive to external conditioning. Critical thinking can be seen as the first line of defense against misleading information. However, little is known about the consequences of media use on the critical thinking skills of tweens. In this study, we evaluated the effects of problematic smartphone use on the various stages of critical thinking, comparing high and low tween smartphone users. The results confirm the main hypothesis, namely, that problematic smartphone use is related to the ability to think critically. There was a significant difference between high and low users in the third phase of critical thinking: evaluation of sources.

## 1. Introduction

In contemporary society, media have become an integral part of every person’s daily life, assuming the characteristic of pervasiveness: even if an individual decides not to interact actively and intentionally with such devices, they can be unintentionally exposed to the information therein reported [1]. We are constantly overwhelmed by a high flow of available, real-time information [2]. Contrary to other digital devices such as computers that need a workstation for use, smartphones are easily accessible without time and space limitations [3]. This renders them extremely convenient: the distribution of smartphones in the world has reached 85%, and 6.8 million people possess at least one smartphone they use assiduously [4]. However, smartphone problematic use, defined as an inability to regulate one’s use of the smartphone, can influence people’s physical and mental health [5]. Despite their many advantages, problematic smartphone use can lead to problems in daily life, including sleep disturbances, impaired empathy and emotional intelligence, difficulty adapting, and reduced academic and work performance [6,7,8]. Negative effects also affect cognition. Several studies have associated the use of smartphones with reduced attentive and mnemonic capabilities [9], increased impulsivity, and deficits in inhibitory control [10,11], as well as worse performance in reasoning tasks [12]. Problematic smartphone use can supplant thought processes and induce cognitive avarice [13]. In fact, people who use smartphones do not devote mental efforts to carrying out cognitive activities, as there are several applications that integrate a wide range of them [14]. Some research has shown how these effects are amplified in younger-aged individuals, such as children and adolescents [15,16]. This is worrying considering that the use of digital technologies with reference to smartphones has increased in tweens, i.e., 9–13-year-olds [17]. On the other hand, in the online world, anyone can propose their own truth, and this leads to greater difficulty in tracing reliable sources [18]. Researchers have conceptualized fake news in different ways while attributing it almost the same meaning. McGonagle [19] has described fake news as deliberately constructed information that is spread with the aim of disinforming and inducing individuals, as well as in error, also to passively accept falsified information. Consistent with this view, the authors Duffy et al. [20] have classified fake news as any false and misleading information that is offered to users as legitimate news. The ambiguous and misleading information to which people are exposed through smartphones can influence their judgments and worldviews [21]. It is important, therefore, to learn to discern data that are not supported by empirical evidence by interfacing with information critically to be able to make independent judgments by resisting undue persuasion [22]. Today, public opinion is more powerfully influenced by personal beliefs and popular opinions than by objective facts and evidence [23]. In a reality where the information conveyed is not always reliable, the ability to think critically supports people in their effort to maintain beliefs that are consistent with the available evidence [24]. Critical thinking (CT) is used whenever effective grounds are sought to justify one’s point of view [25,26]. Ennis [27] identified the five hierarchically ordered phases of critical thinking: problem clarification, analysis, evaluation of credibility and agreement between sources, inference, and self-monitoring of thought processes.

### 1.1. Problematic Smartphone Use and Critical Thinking

One of the theories that explains CT is the dual process model. In CT, there are two distinct systems of reasoning: System 1, characterized by automatic and intuitive mental processes; System 2, characterized by controlled and deliberate processes [28]. System 1, based on heuristic information processing processes, takes place without the conscious control of the individual who makes quick judgments and decisions based on previous knowledge and beliefs. System 2, based on analytical information processing processes, takes place under the deliberate control of the individual [29]. This latter system is slower than the former and requires more cognitive resources but allows the processing of information in a logical way, therefore adhering to the standards of CT [30]. A wide range of studies have shown that often, when reasoning in situations requiring in-depth processing of information, the dominant mode of thinking is the heuristic system which, being negatively associated with CT, leads to the formulation of wrong judgments and opinions [31,32,33]. The risk of accepting information based on heuristics rather than evidence is even more amplified by exposure to media content [34], which leads to a strengthening of this mode of reasoning. The ability to share and spread information through smartphones in an easy and immediate way allows the public to contribute to the progression of news [35]. This leads not only to an increase in the probability of spreading fake news if users are not vigilant about the reliability of sources [36], but also potentially increases the cognitive, emotional, and behavioral influence of family members, friends, and acquaintances who update themselves on essential issues [37]. The media encourage heuristic processing also through the phenomenon of “news snacking”. This expression refers to the occasional and superficial consumption of news, without devoting much time to understanding and evaluating the content. The headlines, the highlighted keywords, and the evocative images all draw the attention of users who focus on these salient elements of the news, leaving out the deep message that wants to be communicated [38]. This contributes to the construction of an overall idea of the content without a thorough understanding. A further threat to CT are the algorithms widely used by social media [39]. The information is filtered according to the opinions and preferences of users, so the same content is continually proposed, consistent with their views. In the absence of contrasting and diversified points of view, reasoning is limited [40]. On the other hand, users themselves tend to choose to stay in contact with people that share the same ideas as they do [41] and, through smartphones, have the possibility of unfriending or unfollowing those who think differently [42]. This phenomenon has been called the “bubble effect”, as it leads to the consumption of only information that is in line with their perspectives [43]. Only when the heuristic response is blocked and replaced by CT can we make an accurate choice of the information we use to form opinions or make decisions [44]. Therefore, CT plays an essential role in media literacy, which implies education in the use of media and the ability to critically evaluate the contents which one comes into contact with [45].

### 1.2. Problematic Smartphone Use and Tweens

Tweens refers to the age group including 9–13 years that represents the intermediate position between being a child and being a teenager. This phase of development is usually characterized by many ambiguities due to the search for one’s own identity and the desire for emancipation and independence [46]. At the cognitive level, there is a shift from concrete to abstract thinking, although, generally, higher executive abilities are not yet fully developed [47]. This is the period of life in which you usually get the first smartphone at an average age of 10 years, and in which you reach the peak of greatest use [48]. Recent studies have shown that tweens interact with their smartphones constantly, with 88% of them checking their phones at least once an hour [49,50]. Moreover, tweens tend to spend a lot of time away from home and often escape parental control and compliance with smartphone usage regulations [51]. Through these devices, they carry out several activities, including browsing the Internet, using social networking, and searching for news [52,53]. Tweens in particular use smartphones as the main reference for finding out information of all kinds [54,55], despite research having found that this age group is particularly susceptible to unreliable news sources [56].

As explained above, problematic smartphone use encourages the use of heuristic reasoning incompatible with CT. However, there is little research in linking media news consumption by tween users with their ability to think critically about real-life news. In a context in which the conditioning of media has become inevitable, it is important to clarify how CT skills are affected. The debate about the use of smartphones by tweens has raised concerns about the psychological and social consequences also because inappropriate habits developed during preadolescence tend to lead to problems in adulthood.

### 1.3. Aims of the Present Work

The main objective of this study was to examine how smartphone usage affects critical thinking (CT) skills in adolescents. Specifically, the first hypothesis was that adolescents who use smartphones frequently would exhibit lower levels of CT performance than those who use them less frequently. The second objective was to identify which stages of CT (clarification, analysis, evaluation, inference, and self-monitoring) showed the most significant differences between high and low smartphone users. Lastly, the third objective was to determine if there were any significant differences in CT proficiency between high and low smartphone users in two distinct problem categories, namely, social and mathematical problems.

## 2. Materials and Methods

### 2.1. Participants

Participants were chosen from a group of 968 tweens (496 girls and 472 boys) attending 45 different classes of public middle schools in Sicily, a region located in southern Italy. All participants were between 11 and 13 years old (M = 12.5, DS = 1.98) and of Italian nationality. Recruitment was carried out by administering a questionnaire that enabled the identification of high and low users of smartphones. Informed consent was obtained from all participants and their parents. To ensure the reliability of the media exposure data, parents completed a version of the questionnaire that was adapted for hetero-evaluation of media exposure. The final sample comprised 100 subjects, 50 of whom were high smartphone users (averaging 9.23 ± 1.23 h of use per week), and 50 of whom were low smartphone users (averaging 3.11 ± 1.27 h of use per week).

### 2.2. Procedure

The investigation was carried out following the principles stated in the Declaration of Helsinki. Each participant voluntarily agreed to participate in this study, and only after tweens’ parents were asked for written informed consent and it was received did the investigation begin. The sample was recruited through the administration of a preliminary questionnaire lasting about 15 min aimed at detecting media usage habits. While parents accompanied their children to school, teachers asked them to fill out the same questionnaire adapted for hetero-evaluation. The results of both administrations were compared, and 23 participants were excluded based on the discrepancy of hours declared by parents and children. The highest and lowest smartphone users were then selected (50 participants with the highest results and 50 with the lowest results). In subsequent days, meetings in presence were planned. The participants were invited to a classroom within the school premises where they underwent a 60 min CT assessment test. The testing was conducted in the morning between 9:00 a.m. and 12:00 p.m. The examiner provided each adolescent with the necessary materials containing two passages and corresponding questions. The tweens were instructed to read the first passage attentively and answer the related questions after finishing the reading. This process was then repeated for the second passage.

### 2.3. Measurements

In this study, two versions of the same questionnaire were used specially adapted for self-assessment and hetero-assessment smartphone use of tweens. Performance tests were used to assess CT skills.

#### 2.3.1. Measurement of Smartphone Use

To assess the intensity of smartphone use, a questionnaire was adapted consisting of 6 items with responses based on a 4-point Likert scale, from 1 (never) to 4 (always). From internal consistency analysis, Cronbach’s alpha was α = 0.81. The aim of this measurement was to identify the highest and lowest smartphone users; therefore, the final sample consisted of the highest and lowest scores for the number of hours spent using the smartphone each day. In order to further confirm the sample, the same questionnaire was adapted and administered to the parents of the preteens. The answers to the questions of the two self-evaluation and external evaluation questionnaires were then compared to make an even more precise distinction between high and low smartphone users. Table 1 and Table 2 show Smartphone Use Questionnaire adapted for auto and hetero-evaluation.

#### 2.3.2. Measurement of Critical Thinking

A CT assessment test was customized, incorporating the 5 phases identified earlier in this research paradigm. Two tracks, namely, track 1 and track 2, were designed for the purpose. Track 1 presented a social dilemma associated with the selection of lower secondary school, while track 2 contained a mathematical problem, which was calibrated based on the subject knowledge of the participants. Both tracks consisted of relevant information crucial for solving the problem, as well as misleading, redundant, or inconsistent information. Each track concluded with a question that encapsulated the reasoning objective. Participants were instructed to read the tracks carefully and attempt to solve the given problem by answering a set of questions. To be precise, three questions were presented for each of the five CT phases (Table 3 and Table 4). An example question related to the inferential phase was “on what evidence do you base your result?”. Also, in this case, a score of 1 was assigned to each correct answer related to questions proposed for the evaluation of the specific phases of CT. On the contrary, a score of 0 was assigned to each incorrect answer related to the questions proposed for the evaluation of the specific phases of CT. Therefore, also in this case, for each CT phase a total score was assigned between 0 (subject answered all questions incorrectly) and 3 (subject answered all questions correctly). From internal consistency analysis, Cronbach’s alpha was α = 0.83. The two texts were presented in a randomized order to ensure counterbalance. The tracks included in the study are as follows:

Track 1—Social problem

Giuseppe has to decide which lower secondary school to attend. His friend Anna told him that the Technical Institute on Via Gramsci has a well-equipped computer lab, and she thinks Giuseppe would do well there. In the mailbox, Giuseppe, who often observes his father growing a small vegetable garden and sometimes he helps him with enthusiasm, finds a flyer that summarizes a scientific high school where a very good philosophy professor teaches. Technical Institute students were invited to a meeting to talk about their own school, organized by a group of parents with purple t-shirts, including Giuseppe’s parents; professors are considered very open to dialogue and the fact that there is a well-stocked sandwich shop near the school is jokingly praised. Leafing through the newspaper, Giuseppe finds an interview with a linguistic high school principal, where the principal accentuates the vanguard of multimedia facilities and the presence of a library full of texts on biology and chemistry. Francesco wants to go to school with Giuseppe and he would like to be a chemical expert. So, he tells Giuseppe that his brother (who has good grades and likes his school) does not get on well at the Agricultural Institute. Anna, who likes Giuseppe, wants to enroll in the technical institute of Via Gramsci, although it is not at the technological forefront. Giuseppe is a lazy reader and does not like to travel. He likes playing with electric trains, but one day, he got an electric shock and since then he has had little to do with electricity. He is a great goalkeeper and hates computers. On Sunday he goes to visit his grandparents in the countryside where they own a farmhouse surrounded by land that they have always cultivated. He enjoys getting on the tractor driven by his grandfather who shows him the farm. A few years earlier, Giuseppe’s aunt gave him a game for little chemists, which Giuseppe used about three times. If you were Giuseppe, which school would you choose?

Track n. 2 (mathematical problem)

One day Wizard Zurlì meets King Alessandro and asks him: “Sire, how old are your 3 children?” And the King answered: “Do you not know? Didn’t you ask anyone on the way here?” Then, the wizard replies: “My dear King, you should know that first I went to the old forgetful Smemo, who rarely visits these lands. He told me about your 3 children, saying that one son was younger than Teo, one was older, and one was probably the same age as Teo. I don’t exactly remember your children’s ages. So, I went to see your uncle Leonardo, who offered me some warm, fragrant tea in his house under the big tree”. He told me he was so proud of his three grandchildren, but being a strict man, the age of each never disclosed, only the product of their ages: 36. I told him “Dear friend, it’s not enough for me”. He replied with a smile.” I know, but I have some other things to do”. So, I picked some strawberries in the woods. Then I went to Ranco, who served me a cup of coffee. He told me that your name is Beniamino, that your children have grown up and gone to the lands to the north. He asked me if I liked the beer, and he went to sleep. Then I met Ben, great mushroom hunter and your dear companion of youth. We walked around, I told him about my journey and my encounter with your uncle Leonardo, and of my deep curiosity. He told me that the sum of your 3 children’s ages is 13. I started thinking, but I realized that I couldn’t determine the exact solution. When I was about to rail against him, I realized he had gone into the woods “I’m too distracted, I said to myself”. The King, amused by the story of the wizard, says: “My dear Zurlì, you’re so distracted. But now that you’re here, you’ll know the truth. I won’t tell you the solution, but I’ll give you some other information. Know therefore that the youngest are twins”. Zurlì, satisfied, finally finds the 3 ages. What do you think these are?

### 2.4. Statistical Analysis

The statistical analysis was performed using SPSS 24.0 software (SPSS Inc., Chicago, IL, USA). The measurement parameters were the mean of correct responses (CRs) for each critical thinking (CT) phase of every participant (high and low smartphone users) in each experimental condition (social and mathematical context). Descriptive statistics were provided for each variable. Regarding the first hypothesis, which states that adolescents who frequently use smartphones would exhibit lower levels of CT performance than those who use them less frequently, a three-factor split–split plot design was employed. This design consisted of two within-subjects factors and one between-subjects factor: 2 (groups: high vs. low smartphone users) X 2 (context: social vs. mathematical) X 5 (critical thinking phases: clarification, analysis, evaluation, inference, and self-monitoring), and for the first hypothesis, we analyzed the Group factor. To identify which stages of CT showed the most significant differences between high and low smartphone users, we considered the Phase factor and the interaction Group X Phase factor as the second objective. Lastly, to determine if there were any significant differences in CT in two distinct problem categories, namely, social and mathematical problems, we considered the Context factor and the interactions with it. An alpha level of 0.05 was set for all statistical tests. Whenever significant effects were observed, the effect size of the test was reported. We used partial eta squared p2 for ANOVA and Cohen’s d Effect Size for the t-test. Additionally, probability values for repeated measures were adjusted using the Greenhouse–Geisser adjustment for non-sphericity.

## 3. Results

Table 5 shows the means (Ms) and standard deviations (SDs) related to correct performance on the five phases of CT in low and high smartphone users and in social and mathematical contexts.

High smartphone users compared to low users show significant differences in the application of CT (F (1, 98) = 8.6, *p* < 0.001). Moreover, the phase factor shows significant effects (F (4, 362) = 14.508, *p* < 0.001). Despite high smartphone users showing lower scores at each stage, the most noticeable difference is in the third stage. Figure 1 and Figure 2 show a sharp drop in the performance of high smartphone users in the evaluation of the source phase. Additionally, the interaction phases X Groups shows a significant effect (F (4, 361) = 5.73, *p* < 0.01). These data show that subjects reason differently in different phases of CT, depending on whether they are high or low media users. Instead, the context variable does not show significant differences. The high and low users do not differ in the indices of CT when answering the leading questions related to a social or mathematical context (F (1, 167) = 0.123, *p* = 0.72). Nevertheless, the interaction context X phases presents highly significant differences (F (4, 88) = 11.361, *p* < 0.001). If we consider the self-monitoring phase, we can see that in the social context, it is correctly performed by several subjects, which is in contrast to what happens in the mathematical context. This may indicate the high relevance that some stages take in one context rather than another.

## 4. Discussion

In the present study, the CT skills of high and low smartphone users in adolescence were evaluated. The results confirmed the main hypothesis that problematic smartphone use is related to CT. Participants with a low exposure seem to be more reflective than participants with higher fruition of smartphones. This is consistent with the findings of Pérez-Rodríguez et al. [34] showing how exposure to media content reinforces heuristic-based reasoning, which is negatively associated with CT [30]. The data obtained further support our hypothesis, particularly as high smartphone users exhibited the poorest performance across all CT phases, including clarification, analysis, inference, and self-monitoring, with the most significant discrepancy observed in the most challenging phase, source evaluation. This finding is in line with previous research, which suggests that errors during source evaluation increase the risk of making incorrect judgments due to unreliable information [56]. Another objective of this study was to examine whether there were any significant differences in critical thinking (CT) abilities between high and low smartphone users regarding two distinct problem types: social and mathematical. The study found that CT was better exercised in the social context than in the mathematical context. This is likely because, in more practical contexts, such as the social context, there are fewer cognitive resources involved, allowing for more resources to be allocated toward logical reasoning. Our results could also be explained based on cognitive load theory (CLT) [57]. As suggested by Fabio and Suriano [35], a high level of media exposure decreases working memory performance, leaving few cognitive resources available for CT.

Previous studies have shown that media use negatively affects the ability to think critically [58,59,60,61,62], but the impact of problematic smartphone use on Italian tweens has been little-studied. This study provides a novel insight into the impact of smartphone usage on the critical thinking (CT) skills of Italian tweens, which has not been previously explored in existing research that mainly examined CT development in children and adults, excluding tweens [63,64]. A unique aspect of this study is that it examines CT differences by analyzing specific phases (clarification, analysis, evaluation, inference, and self-monitoring) between higher and lower smartphone users, unlike previous research that only focused on studying CT in individuals with smartphone addiction [65,66]. We are aware of only one study that evaluated this relationship in the Chinese population [36]. Our work is a contribution to the generalization of the results as sociocultural characteristics shape the values and beliefs of people, which in turn affect behavior toward the news [67].

### 4.1. Limitations of the Study

Among the limitations of this study, we evaluated the effects of the intensity of daily use of the smartphone, neglecting the effects of using other media on CT. The effects of media use are multifactorial and depend on the type of media used and the amount and intensity of use, as well as the characteristics of the individual who uses them. Another limitation could be the statistical analysis. A structural equation model for contrast models would introduce a more comprehensive understanding of the relationships between the variables and the underlying mechanisms that explain the result.

Another limitation could be the neglect of the measurement of individual characteristics and contextual factors that could mediate the effects of media use on CT.

### 4.2. Future Prospects

Future research should analyze and compare the effects of using additional digital devices or platforms, as well as consider intervening variables such as digital alphabetization or personality traits to outline a clearer picture of how CT phases are affected. More in depth, based on the limitations stated in the study, there are several future prospects for research. The first one, as we said, may be investigating the effects of using other media on CT: future studies can evaluate the effects of different types of media (e.g., TV, video games, and social media) on CT and compare them with the effects of smartphone use. This can provide a more comprehensive understanding of the impact of media use on cognitive abilities. The second could be to examine the role of individual characteristics: researchers can investigate how individual characteristics (e.g., age, gender, and personality traits) influence the relationship between media use and CT. This can help identify vulnerable populations who may be more susceptible to the negative effects of media use. With reference to social factors, future studies can explore the impact of contextual factors (e.g., socio-economic status, cultural background, and education) on the relationship between media use and CT. This can help identify the cultural and societal factors that may influence the effects of media use on cognitive abilities. The fourth could be the use of advanced statistical analysis; researchers can use advanced statistical models such as structural equation models to gain a more comprehensive understanding of the complex relationships between the variables and underlying mechanisms that explain the results. This can help provide more precise estimates of the effects of media use on CT.

## 5. Conclusions

The current paper yields two key findings: (1) individuals who use smartphones less problematically exhibit greater levels of reflection compared to those who use smartphones more frequently; (2) critical thinking about social issues appears to be easier than critical thinking about mathematical problems. Considering the importance of critical thinking as a valuable cognitive process in problem solving [68,69,70], educational institutions should implement targeted strategies to foster the development of this skill. Developing critical thinking abilities can help individuals overcome biases and thinking errors, leading to a more objective understanding of reality.

## Figures and Tables

**Figure 1 children-10-00698-f001:**
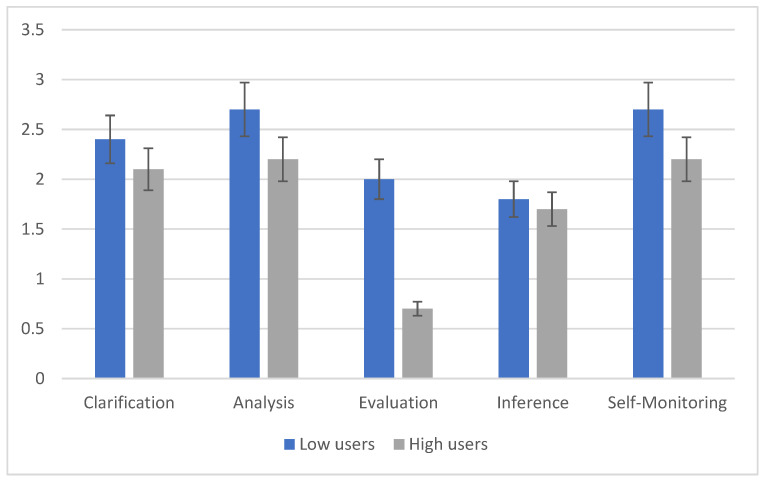
Means scores related to the phases of critical thinking of the social problem.

**Figure 2 children-10-00698-f002:**
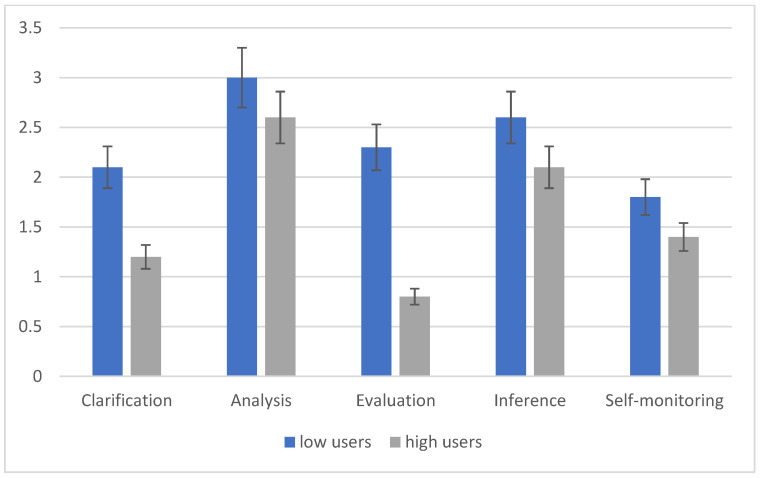
Means scores related to the phases of critical thinking of the mathematical problem.

**Table 1 children-10-00698-t001:** Smartphone Use Questionnaire adapted for auto evaluation.

1. How much time per day do you spend using these technology devices? *	☐1 ☐2 ☐3 ☐4
2. I feel lost without my smartphone	☐1 ☐2 ☐3 ☐4
3. I spend more time on my smartphone than I intend to	☐1 ☐2 ☐3 ☐4
4. I check my smartphone frequently throughout the day	☐1 ☐2 ☐3 ☐4
5. I use my smartphone during meals or other social events	☐1 ☐2 ☐3 ☐4
6. I have trouble putting my smartphone down, even when I know I should	☐1 ☐2 ☐3 ☐4

* 1 (never to 1 h), 2 (1 h to 3 h), 3 (3 to 6 h), 4 (more than 6 h).

**Table 2 children-10-00698-t002:** Smartphone Use Questionnaire adapted for hetero-evaluation.

1. How much time per day does your child spend using these technology devices? *	☐1 ☐2 ☐3 ☐4
2. Your child feels lost without his smartphone	☐1 ☐2 ☐3 ☐4
3. Your child spends more time on his smartphone than he intends to	☐1 ☐2 ☐3 ☐4
4. Your child checks his smartphone frequently throughout the day	☐1 ☐2 ☐3 ☐4
5. Your child uses his smartphone during meals or other social events	☐1 ☐2 ☐3 ☐4
6. He has trouble putting his smartphone down, even when he knows he should	☐1 ☐2 ☐3 ☐4

* 1 (never to 1 h), 2 (1 h to 3 h), 3 (3 to 6 h), 4 (more than 6 h).

**Table 3 children-10-00698-t003:** Scoring for each question of the social problem.

	First Question	Second Question	Third Question	Replies	Scores
**Clarification**	What kind of problem is this?	What is the central point of the problem?	Can you think of similar problems to this?	1st It is a choice.	1
				2nd The choice of which lower secondary school to attend.	1
				3rd To choose the best sport.	1
**Analysis**	What are the parts of this problem?	What are the links among the information you have read?	What are the links with the final question?	1st Giuseppe has to choose lower secondary school. His friends, his family, and the news he reads may influence him.	1
				2nd Interests, friends, and news.	1
				3rd How these three factors affect the choice.	1
**Evaluation**	What information can you believe and what information is not credible?	Is there any unhelpful information that is not needed to solve the problem?	Does all the information contained in the problem agree or contradict itself?	1st I can’t believe Francesco, because he tells a lie to convince Giuseppe to attend his own school.	1
				2nd The color of the parents’ T-shirt who organized the meeting to which technical students were invited is not relevant.	1
				3rd With reference to the Via Gramsci Technical Institute, first it is said that it has a good computer lab and later that it is not at the technological forefront.	1
**Inference**	What result have you achieved?	What information do you base your result on?	Could you think of something different as a solution?	1st Maybe Agricultural Institute.	1
				2nd The fact that he’s excited when he helps dad and grandpa work in the garden.	1
				3rd Yes. I can think about a scientific school.	1
**Self-monitoring**	Was it difficult to arrive at the solution?	Did you check if you thought correctly?	Have you found something that does not convince you?	1st Yes, a little bit, maybe too much information.	1
				2nd Maybe I made a mistake in remembering all the names, I can check again if I have time.	1
				3rd Yes. For example, the scientific school question.	1

**Table 4 children-10-00698-t004:** Scoring for each question of the mathematical problem.

	First Question	Second Question	Third Question	Replies	Scores
**Clarification**	What kind of problem is this?	What is the central point of the problem?	Can you think of similar problems to this?	1st It is an algebraic problem.	1
				2nd The calculation of ages of the king’s children.	1
				3rd The calculation of three brother’s ages.	1
**Analysis**	What are the parts of this problem?	What are the links among the information you have read?	What are the links with the final question?	1st Wizard Zurlì wants to find out the ages of the king’s children. People he meets give him information.	1
				2nd To relate the information given by uncle Leonardo, Ben, and King to the conclusion.	1
				3rd How this information can be useful to find the solution.	1
**Evaluation**	What information can you believe and what information is not credible?	Is there any unhelpful information that is not needed to solve the problem?	Does all the information contained in the problem agree or contradict itself?	1st I can’t believe Ranco because he says the king’s name is Beniamino.	1
				2nd It is not relevant where Smemo lives.	1
				3rd The King’s name is Alessandro, but later Ranco said Beniamino.	1
**Inference**	What result have you achieved?	What information do you base your result on?	Could you think of something different as a solution?	1st The ages of the king’s children are: 2, 2, 9.	1
				2nd About calculations and the fact that the two young brothers are twins.	1
				3rd Yes, I could have thought 1, 6, 6, but the twins are younger than the older bother.	1
**Self-monitoring**	Was it difficult to arrive at the solution?	Did you check if you thought correctly?	Have you found something that does not convince you?	1st Yes, a little bit, maybe too much information.	1
				2nd Maybe I made a mistake in remembering all the. names, I can check again if I have time.	1
				3rd Yes. For example, Ranco’s statement.	1

**Table 5 children-10-00698-t005:** Descriptive statistics of the mean of correct replies and standard deviations in each phase of critical thinking.

Groups		Social	Mathematical
Low smartphone users	Clarification	2.41 (±0.245)	2.16 (±0.261)
	Analysis	2.75 (±0.212)	3.00 (±0.159)
	Evaluation	2.00 (±0.277)	2.33 (±0.198)
	Inference	1.83 (±0.391)	2.67 (±0.264)
	Self-monitoring	2.75 (±0.250)	1.83 (±0.158)
High smartphone users	Clarification	2.08 (±0.245)	1.25 (±0.261)
	Analysis	2.16 (±0.212)	2.66 (±0.159)
	Evaluation	0.750 (±0.277)	0.833 (±0.198)
	Inference	1.66 (±0.391)	2.16 (±0.264)
	Self-monitoring	2.25 (±0.250)	1.41 (±0.158)

## Data Availability

Data are available on request to each of the authors.

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
