# Peer review of "The Influence of Smartphone Use on Tweens’ Capacity for Complex Critical Thinking"

_children, 2023, doi:10.3390/children10040698_

Round 1
Reviewer 1 Report
n the introduction the authors hould delete the official requrements for this section:
Suggestion:
Delete the first 8 rows (22-29, and first half of row 30):
The introduction should briefly place the study in a broad context and highlight why it is important. It should define the purpose of the work and its significance. The current state of the research field should be carefully reviewed, and key publications cited. Please highlight controversial and diverging hypotheses when necessary. Finally, briefly mention the main aim of the work and highlight the principal conclusions. As far as possible, please keep the introduction comprehensible to scientists outside your particular field of research. References should be numbered in order of appearance and indicated by a numeral or numerals in square brackets—e.g., [1] or [2,3], or [4–6]. See the end of the document for further details on references.
and start only with the phrase:
In contemporary society, media have become an ....see row 30
The research is well designed, and covers an important subject. the number of sample is rather imoressive. The testing is simple but well documented, and conclusions are summary.
Author Response
In the introduction the authors should delete the official requirements for this section:
Suggestion:
Delete the first 8 rows (22-29, and first half of row 30):
The introduction should briefly place the study in a broad context and highlight why it is important. It should define the purpose of the work and its significance. The current state of the research field should be carefully reviewed, and key publications cited.Please highlight controversial and diverging hypotheses when necessary. Finally, briefly mention the main aim of the work and highlight the principal conclusions. As far as possible, please keep the introduction comprehensible to scientists outside your particular field of research. References should be numbered in order of appearance and indicated by a numeral or numerals in square brackets—e.g., [1] or [2,3], or [4–6]. See the end of the document for further details on references.
and start only with the phrase:
In contemporary society, media have become an ....see row 30
Reply
Thank you. We deleted this part
The research is well designed, and covers an important subject. the number of sample is rather imoressive. The testing is simple but well documented, and conclusions are summary.
Reply
Thank you.

Reviewer 2 Report
This is an interesting study on the association between smartphone use and critical thinking ability. The study is well-conducted and I agree that it may contribute well to the literature. However, I have several comments that need to be addressed:
1. It is important for the authors to proofread their paper. For example, in the introduction, they should remove the instruction on how to write the introduction. It is still there (i.e., The introduction should briefly place the study in a broad context and highlight why it is important. It should define the purpose of the work and its significance ... details on references).
2. In the introduction, I believe that the authors should also review on studies that found evidence demonstrating that smartphones are used to supplant thinking and induce cognitive misery and failures. This will provide another line of literature and theoretical framework to support on how smartphone use may predict lower complex critical thinking. Here are some relevant studies:
Smartphone use and daily cognitive failures: A critical examination using a daily diary approach with objective smartphone measures. (2023). British Journal of Psychology, 114(1), 70-85. The brain in your pocket: Evidence that smartphones are used
to supplant thinking. (2015). Computers in Human Behavior, 48, 473– 480. 3. More information should be provided on how the hereto evaluation of media exposure was administered. The 6 items of measurement of smartphone use should also be elaborated in the method section. 4. The reliability of the scales in the study should be reported. 5. Given that the study is a cross-sectional study, there is a need for the authors to tone down the discussion and avoid the use of causal languages when interpreting the findings.
Author Response
REV 2
This is an interesting study on the association between smartphone use and critical thinking ability. The study is well-conducted and I agree that it may contribute well to the literature. However, I have several comments that need to be addressed:
1.It is important for the authors to proofread their paper. For example, in the introduction, they should remove the instruction on how to write the introduction. It is still there (i.e., The introduction should briefly place the study in a broad context and highlight why it is important. It should define the purpose of the work and its significance ... details on references).
Reply
Thank you. We deleted this part
- In the introduction, I believe that the authors should also review on studies that found evidence demonstrating that smartphones are used to supplant thinking and induce cognitive misery and failures. This will provide another line of literature and theoretical framework to support on how smartphone use may predict lower complex critical thinking. Here are some relevant studies:
Smartphone use and daily cognitive failures: A critical examination using a daily diary approach with objective smartphone measures. (2023). British Journal of Psychology, 114(1), 70-85. The brain in your pocket: Evidence that smartphones are used to supplant thinking. (2015). Computers in Human Behavior, 48, 473– 480.
Reply
Thank you for your suggestion! We included the works you suggested as follows:
Despite their many advantages, problematic smartphone use can lead to problems in daily life, including sleep disturbances, impaired empathy and emotional intelligence, difficulty adapting, and reduced academic and work performance [6-8]. Negative effects also affect cognition. Several studies have associated the use of smartphones with reduced attentive and mnemonic capabilities (Throuvala et al., 2021), increased impulsivity, and deficits in inhibitory control (Chen et al., 2016; Hartanto & Yang, 2016), as well as worse performance in reasoning tasks (Pluck et al., 2020). Problematic smartphone use can supplant thought processes and induce cognitive avarice (Barr et al., 2015). In fact, people who use smartphones do not devote mental efforts to carrying out cognitive activities, as there are several applications that integrate a wide range of them (Hartanto et al., 2023). Some research has shown how these effects are amplified in younger-aged individuals, such as children and adolescents [9, 10].
- More information should be provided on how the hereto evaluation of media exposure was administered. The 6 items of measurement of smartphone use should also be elaborated in the method section.
Reply
Thank you. We added the full scale of the 6 items of measurement of smartphone both with reference to auto and hetero evaluations in the Measurements section. We provided also the information on the hetero evaluation in the procedure section.
- The reliability of the scales in the study should be reported.
Reply
Thank you. We added the reliability.
- Given that the study is a cross-sectional study, there is a need for the authors to tone down the discussion and avoid the use of causal languages when interpreting the findings.
Reply
Thank you. We softened the causal relationship.

Reviewer 3 Report
It seems to be a good manuscript. However, I invite the authors to make the revised considerations to be published. Thank you.

Author Response
Replies to referee 3
The Influence of Smartphone Use on Tweens’ Capacity for Complex Critical Thinking (children-2252950)
Initial comment: In general, the manuscript seems to me an appropiate paper to be published. The authors made a sufficiently consistent introduction and the analysis methods are adequate. The discussion responded to a greater or lesser extent to the research problems.
Reply
Thank you
However, I would like to make some clarifications that would improve the quality of the article.
Please answer each of the sections separately:
Abstract: The abstract is according to the journals's guidlines.
Keywords: It is recommended to introduce a maximum of five keywords as well as to order them alphabetically.
Reply
Thank you. We ordered them alphabetically and we leave 3 keywords (green highlighted)
- Introduction: The introduction is appropriate and summarizes the objective of the study. However, the hypotheses or research problems must be established at the end of the section instead of the three general objectives
- Reply
Thank you. We added and ordered them alphabetically and we put 3 keywords (green highlighted)
- Material and methods:
Participants: It must be indicated how the sample was selected for the investigation. Informed consent must be mentioned.
Measurement: A detailed description of the questionnaires should be mentioned.
Statistical Analysis: Statistical analysis should be written more clearly relating it to the research objectives or intended results.
- Reply
Thank you. We added how we selected the sample (green highlighted).
We added the precise instruments used.
We rewrote the statistical analysis following your suggestion.
- Results: The statistical treatment is simple. I would introduce some causal relationship in the form of a structural equation model for contrast models that propose causal relationships between the study variables.
- Reply
Thank you. I would like to acknowledge that due to my limited knowledge and experience with structural equation modeling, I was unable to perform this specific analysis in the current study. However, I have addressed this limitation by discussing the potential implications of not including this analysis in the limitations section of the manuscript. I hope that this disclosure will not negatively impact the overall evaluation of the study
- Discussion: The discussion is correct according to the information provided, although I still think that a better statistical treatment would considerably improve the manuscript.
4.Reply
Yes, it is related to the previous reply
- Conclusions: Appropriate conclusions in line with the results and the discussion although I would lengthen the section. The sections "Limitations of the study" and "Future prospects" must be included.
Minor revisions:
- a) Keywords: “social and mathematical contexts” its not a keyword. Please, reduce the words – Line 19
- We did it
- b) Quotes n 11,12,13 are in different type text.
- done
- c) Revise the whole bibliography
- we revised it

Round 2
Reviewer 1 Report
I accept the corrections made in the introduction.
Author Response
We made all the relevant changes
Reviewer 2 Report
The authors have addressed my comments well.
Author Response
Thank you, we made all the changes